# Effect of Biomimetic Surface Geometry, Soil Texture, and Soil Moisture Content on the Drag Force of Soil-Touching Parts

Abouelnadar El. Salem [1,2], Hongchang Wang [1], Yuan Gao [1], Xiantao Zha [1], Mohamed Anwer Abdeen [1,3] and Guozhong Zhang [1,*]

[1] College of Engineering, Huazhong Agricultural University, Wuhan 430070, China; abouelnadar@webmail.hzau.edu.cn (A.E.S.); wanghc84@mail.hzau.edu.cn (H.W.); gaoyuan@mail.hzau.edu.cn (Y.G.); zhaxiantao@webmail.hzau.edu.cn (X.Z.); mohamed.anwer2010@yahoo.com (M.A.A.)
[2] Desert Research Center, Mataria 11753, Egypt
[3] Agricultural Engineering Department, Zagazig University, Zagazig 44519, Egypt
* Correspondence: zhanggz@mail.hzau.edu.cn; Tel.: +86-1867-2783-365

**Abstract:** Soil adhesion is a major problem for agricultural machinery, especially in sticky soils within the plastic range. One promising and practical way to minimize soil–tool adhesion is to modify the surface geometry to one inspired by soil-burrowing animals. In this study, 27 domed discs were fabricated according to an L27 ($3^3$) Taguchi orthogonal array and tested to determine the optimal dimensions of domed surfaces to reduce drag force. The optimized domed disc was tested in a soil bin under different soil conditions (soil texture: silty loam and sandy clay loam; soil moisture content: 23%, 30%, and 37%). All trials included a flat disc (without a dome pattern) as a control. The optimal dimensions of domed surfaces to generate the lowest possible drag force under the present experimental conditions were explored based on signal-to-noise ratio analysis. The optimal levels of control parameters were found at a surface coverage ratio of 60%, dome height of 5 mm, and dome base diameter of 20 mm. Statistics revealed that the dome height-to-diameter ratio and disc coverage ratio are crucial factors that influence the drag force of domed surfaces. In contrast, the dome base diameter had a limited influence on drag force. In all treatments, the drag force of the optimized domed disc was less than that of the flat disc (by about 9% to 25%, according to soil conditions). Accordingly, it can be concluded that adequately designed domed surfaces could significantly reduce the drag force in sticky soil compared to their flat counterparts.

**Keywords:** bionic surface; paddy soil; sliding resistance; Taguchi method

## 1. Introduction

In most soil-engaging parts in tillage and planting tools, the soil slides tangentially along the tool surface. The sliding resistance along the soil–tool interface is determined by three forces that interact: (i) structural resistance to soil particle displacement, (ii) frictional resistance to transfer between individual soil particles, and (iii) resistance produced by soil–tool adhesion [1]. Soil adhesion to the tool surface can cause a change in surface shape and substantially increase the sliding resistance.

Soil adhesion is the force of attraction between soil particles and the surface contacting the soil; this attraction often occurs due to the surface tension and viscosity of the interfacial water film (a few molecules thick) forming on the tool surface [2]. Soil–tool adhesion can have various implications for agricultural operations, including lowering germination rates [3], increasing energy consumption per unit soil operation [4], and restricting the use of no-tillage planting in sticky soils [5]. Another consequence of soil adhesion phenomena is a considerable reduction in the work efficiency of loading and excavation equipment [6].

According to Coulomb's theory, the sliding resistance along the soil–tool interface could be described as a function of the normal load applied to the soil–tool interface, the soil–tool contact area, soil–tool adhesion, and soil–tool friction as per Equation (1) [7],

$$\tau = a\,C + N \tan \varphi \tag{1}$$

Here, $\tau$ is the sliding resistance in the tangential direction, (N), $a$ is the contact area, (cm$^2$), $C$ is the coefficient of adhesion, (N/cm$^2$), $\varphi$ is the soil-metal friction angle (degrees), and $N$ is the normal load to the contact interface (N).

Fountaine (1954) [8] initially proposed Equation (1) based on laboratory experiments concerned with the mechanism of soil-metal adhesion, which was later applied to non-metallic materials such ultra-high molecular weight polyethylene [7] and fiber-glass [9]. As a result, Equation (1) could be used to characterize soil-tool sliding resistance regardless of the material used.

In other words, soil–tool sliding resistance includes two terms, namely, friction and adhesion. However, Srivastava et al. [10] pointed out that it is difficult to distinguish between friction and adhesion. As a result, an apparent coefficient of friction is commonly employed to account for both friction and adhesion effects.

According to Soni and Salokhe [11], many factors impact soil–tool adhesion, including soil texture, soil moisture content, tool material, tool geometry, interfacial conditions, and soil Atterberg constants, particularly the sticky point. The sticky point is the soil moisture content where the soil begins to cling to a foreign item. More precisely, it is the soil paste's moisture content at which the soil particles begin to attach to a polished nickel surface under a shearing speed of 50 mm s$^{-1}$ [12].

To date, a variety of techniques have been used to minimize soil–tool adhesion, including changing the composition of the material used to fabricate soil-engaging parts [9], altering the geometric shape of implements [13], generating mechanical and ultrasonic vibrations [14], increasing the adaptability of the soil-engaging components [15], using polymeric materials to coat soil-engaging parts [7,16], and using a bionic electro-osmosis method [17].

In light of the promising results of surface shape modification in lowering soil adherence to primary and secondary tillage implements under certain soil conditions [18,19], it was a natural step to expand experiments to incorporate a broader range of soil conditions, such as those in paddy fields. Many mechanized agricultural operations are carried out during rice transplanting seasons in Hubei Province, China, such as field flattening, fertilization, and direct seeding. During these operations, a large amount of soil clings to soil-engaging components, limiting their efficiency and effectiveness (Figure 1). In this experiment, an L27 ($3^3$) Taguchi design was employed; the Taguchi design is a frontier in data mining techniques that has gained popularity in engineering applications [20]. The Taguchi technique uses signal-to-noise ratio analysis as a measurable statistic "quantitative analysis tool" to find optimal levels of control parameters that lead to an optimized design [21]. The current research aimed to explore the effects of specific geometric dimensions (disc coverage ratio, dome height-to-diameter ratio, and dome base diameter) on drag force under paddy fields conditions. Furthermore, the drag force of the optimized domed disc was compared to that of the flat disc under different soil conditions.

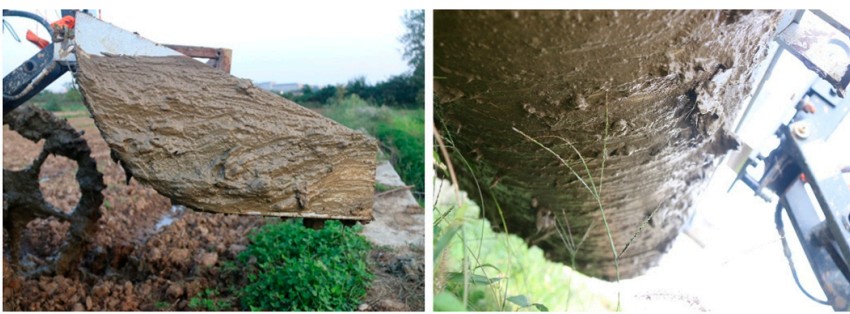

**Figure 1.** Soil–tool adhesion phenomena during the flattening operations in paddy fields.

## 2. Materials and Methods

In this study, the influences of certain geometric features of domed surfaces (disc coverage ratio, i.e., the percentage of the base area occupied by the domes out of the entire area of the disc, dome height-to-diameter ratio, and dome base diameter) on the drag force were investigated. The levels of control parameters were determined based on the recommendations of previous studies and research on the influence of surface shape alterations on soil–tool adhesion [13,22,23] as listed in Table 1. According to the Taguchi design L9 ($3^3$) orthogonal array, preliminary tests were then conducted to validate the rationality of these selections.

**Table 1.** The selected values of control parameter levels.

| Parameter | Symbol | Level Number | | |
|---|---|---|---|---|
| | | 1 | 2 | 3 |
| Disc coverage ratio (CR) (%) | A | 45 | 60 | 75 |
| Dome height-to-diameter ratio (HDR) (%) | B | 12.5 | 25 | 37.5 |
| Dome base diameter (DBD) (mm) | C | 10 | 20 | 30 |

### 2.1. Soil Preparation

Samples of the fertile top layer of soil (0–20 cm) were collected independently from two different sites in Hubei Province, China, namely, a paddy field at the experimental fields of Huazhong Agricultural University in Wuhan city (Longitude: 114°20′49″ E, Latitude: 30°28′26″ N) and a sunflower field in Huangshi city (Longitude: 114°58′47″ E, Latitude: 30°12′02″ N). The texture types of the experimental soils were silty loam and sandy clay loam, respectively. The soil samples were left to dry in open air for about 2 weeks, after which the dry soil was pulverized with a wooden hammer, and the loose material was sieved with a 4 mm mesh sieve. Then, according to Equation (2), the amount of water required to provide the chosen experimental conditions was calculated and added to the sieved soil [24].

$$V_a = \left(SMC_{req} \times W_s\right) - \left(SMC_{ex} \times W_S\right) \tag{2}$$

Here, $V_a$ is the amount of water that must be added to the soil to obtain the required soil moisture content (g), $SMC_{req}$ is the required moisture content (g kg$^{-1}$), $SMC_{ex}$ is the moisture content of the sieved soil (g kg$^{-1}$), and $W_s$ is the weight of the soil (kg).

The Atterberg constants of the experimental soils (plastic limit and liquid limit) were determined in the soil physics laboratories of Huazhong Agricultural University according to standard ASTM D 4318 [25] and are given in Table 2.

**Table 2.** Experimental soil properties.

| Property | Paddy Field | Sunflower Field |
|---|---|---|
| Dry bulk density (kg m$^{-3}$) | 1557 | 1412 |
| Liquid limit (%) | 42 | 36 |
| Plastic limit (%) | 20 | 17 |
| Texture [1] | Silty loam | Sandy clay loam |
| Sand | 22 ± 1.5% | 54 ± 2.5% |
| Silt | 63 ± 1.8% | 25 ± 1.5% |
| Clay | 15 ± 1% | 21 ± 1.1% |

[1] Texture of experimental soils according to the classification of the United States Department of Agriculture (USDA).

### 2.2. Preparation of the Test Discs and Soil Bin

The structural diagram of the test disc is shown in Figure 2. Based on the number of specified parameters and their levels (Table 1), an L27 ($3^3$) Taguchi orthogonal array was adopted to design the current experiment. Accordingly, 27 circular discs with a diameter

of 10 cm were created (Figure 3). The discs were made from acrylonitrile butadiene styrene (ABS), which is widely utilized in 3D printing technology. The test disc geometers were inspired by the micro-convex structure of the dung beetle head [13]. As shown in Figure 3, these discs' bottom surfaces had diverse geometries depending on the different combinations of experimental parameter levels. An appropriate soil bin (55 cm length, 30 cm width, and 30 cm height) was designed to conduct drag force tests (Figure 4b). A sliding part moving over two bars was installed on top of the soil bin, and a telescopic joint was attached to the sliding part responsible for lowering and raising the test discs. A metal sheet was also mounted on the sliding part to level and smooth the soil surface before each trial (Figure 4c). The sliding part without a disc was dragged once to measure the friction forces of the sliding part and pulley system; the results of this run were used to calculate the pure traction force. The pure traction force was calculated to estimate the exact percentage change in drag force produced by modifying the surface shape.

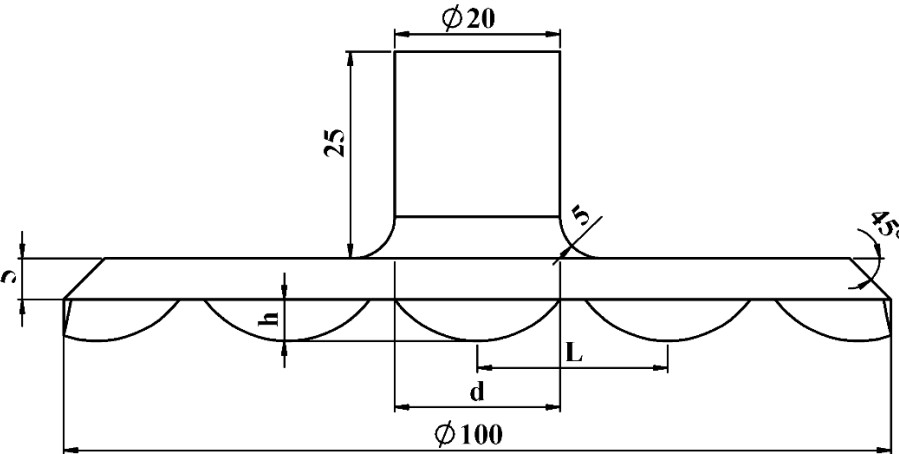

**Figure 2.** Structural diagram of the test disc (dimensions in millimeters).

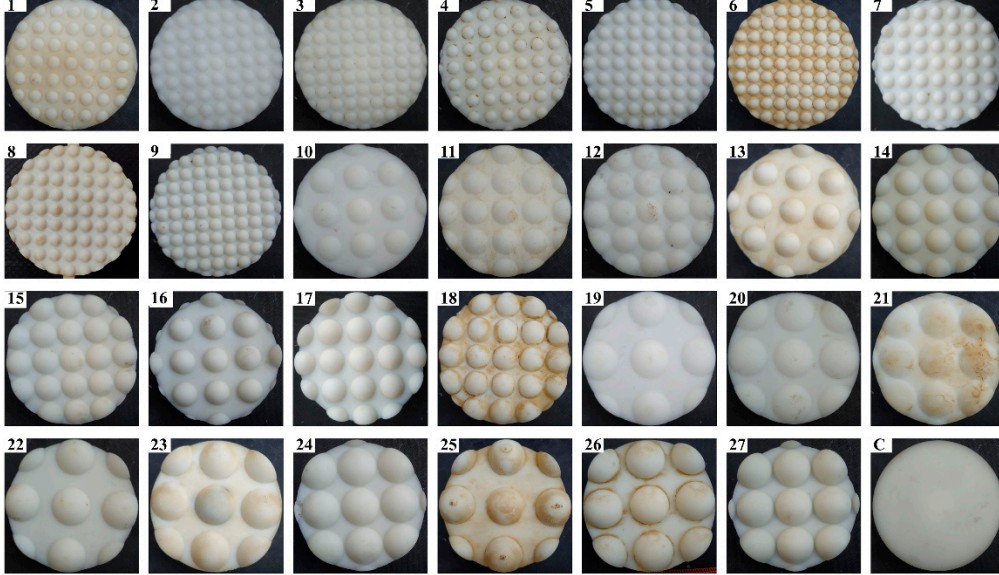

**Figure 3.** The full factorial design of control parameter levels according to the L27 ($3^3$) Taguchi orthogonal array, in addition to the flat disc (C), which uses as a control.

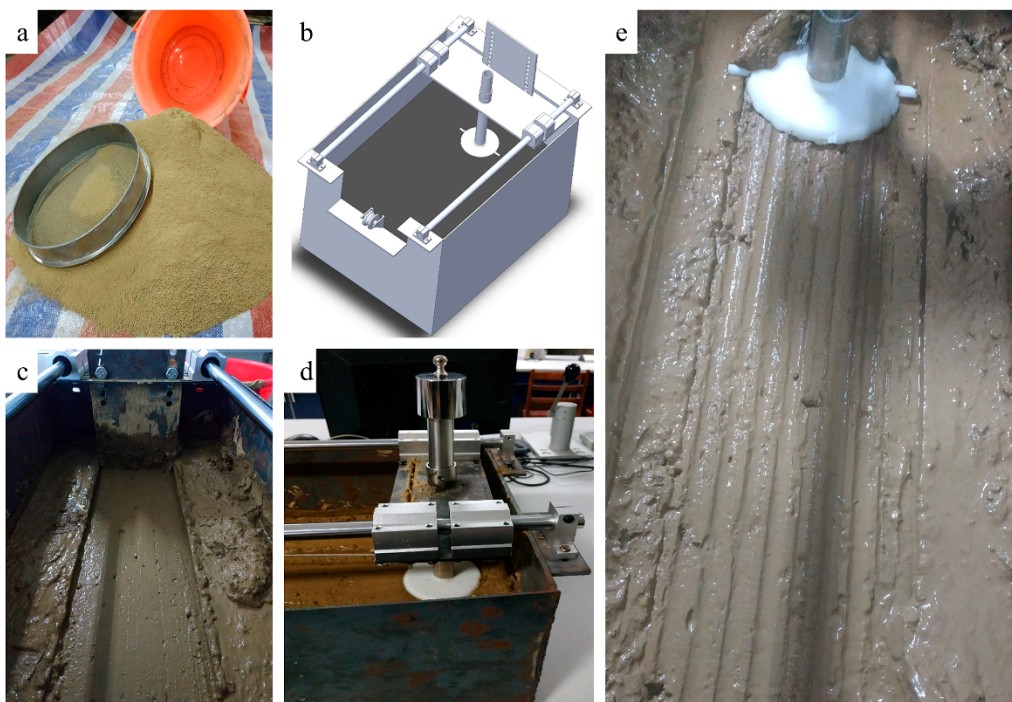

**Figure 4.** Experimental setup for the sliding resistance test: (**a**) the sieved soil; (**b**) creating an appropriate soil bin to test sliding resistance; (**c**) flattening and smoothing of soil surfaces; (**d**) exposing the soil paste to constant normal stress of 1025 N·m$^{-2}$ for one minute; (**e**) pulling the test disc for 40 cm of soil bin length.

### 2.3. Drag Force Test

The drag force of each test disc was measured using a TMS-PRO texture analyzer (food technology corporation of the United States) equipped with a 1000 N load cell, 0.1 N accuracy, and a moving speed range of 0.1–500 mm min$^{-1}$.

Drag force tests were conducted in two consecutive phases, namely, the loading phase and the drag phase. The loading phase intended to sink the entire dome height (column 3 in Table 3) into the soil paste before starting the dragging phase. During the loading phase, all samples were subjected to a sufficient normal load to sink at least the dome height into the soil. The sinkage of each sample under the same load was varied depending on soil moisture content and sample geometrical features. The appropriate normal load and load exposure time were determined through observing and engaging in some exploratory experiments, which revealed that the normal load of 820 gm (1025 N/m$^2$, as all samples had a diameter of 100 mm) was sufficient for all samples in all treatments to sink the entire dome height into the soil within 1 min. After the load exposure duration, the telescopic joint's screw was fastened tightly to keep the test disc at a constant depth during the drag phase. In the drag phase, each disc was pulled for 40 cm of soil-bin length (Figure 4e). The built-in data processing software continuously displayed the traction force during the experiment for real-time observation. Subsequently, the mean traction force was computed individually for each test disc using the data acquired through testing.

**Table 3.** The geometrical specifications of test discs and measured values of drag force.

| Disc No. | Parameter Combinations | h (mm) | n | L (mm) | Drag Force (N) | | | | Δ (%) |
|---|---|---|---|---|---|---|---|---|---|
| | | | | | $R_1$ | $R_2$ | $R_3$ | Mean | |
| Control | - | - | - | - | 9.2 | 9.3 | 8.9 | 9.1 | - |
| 1 | $A_1B_1C_1$ | 1.25 | 45 | 13.0 | 7.8 | 7.6 | 7.4 | 7.6 | 16.5 |
| 2 | $A_2B_1C_1$ | 1.25 | 60 | 11.5 | 7.2 | 7.4 | 7.1 | 7.2 | 20.9 |
| 3 | $A_3B_1C_1$ | 1.25 | 75 | 10.2 | 9.8 | 9.4 | 10 | 9.7 | −6.6 |
| 4 | $A_1B_2C_1$ | 2.5 | 45 | 13.0 | 7 | 6.8 | 6.9 | 6.9 | 24.2 |
| 5 | $A_2B_2C_1$ | 2.5 | 60 | 11.5 | 7.1 | 7.3 | 6.8 | 7 | 23.1 |
| 6 | $A_3B_2C_1$ | 2.5 | 75 | 10.2 | 8.7 | 8.5 | 8.4 | 8.5 | 6.6 |
| 7 | $A_1B_3C_1$ | 3.75 | 45 | 13.0 | 9.8 | 9.4 | 9.2 | 9.5 | −4.4 |
| 8 | $A_2B_3C_1$ | 3.75 | 60 | 11.5 | 8.3 | 8.2 | 8.3 | 8.3 | 8.8 |
| 9 | $A_3B_3C_1$ | 3.75 | 75 | 10.2 | 10.8 | 10.5 | 11.3 | 10.9 | −19.8 |
| 10 | $A_1B_1C_2$ | 2.5 | 11 | 26.0 | 7.5 | 7.3 | 7.2 | 7.3 | 19.8 |
| 11 | $A_2B_1C_2$ | 2.5 | 15 | 23.0 | 7.4 | 7.8 | 8 | 7.7 | 15.4 |
| 12 | $A_3B_1C_2$ | 2.5 | 19 | 20.4 | 9.7 | 9.2 | 9.1 | 9.3 | −2.2 |
| 13 | $A_1B_2C_2$ | 5.0 | 11 | 26.0 | 6.9 | 6.8 | 7.1 | 6.9 | 25.3 |
| 14 | $A_2B_2C_2$ | 5.0 | 15 | 23.0 | 6.6 | 7 | 6.5 | 6.8 | 24.2 |
| 15 | $A_3B_2C_2$ | 5.0 | 19 | 20.4 | 6.8 | 7.3 | 6.9 | 7 | 23.1 |
| 16 | $A_1B_3C_2$ | 7.5 | 11 | 26.0 | 9.5 | 9.3 | 9.2 | 9.3 | −2.2 |
| 17 | $A_2B_3C_2$ | 7.5 | 15 | 23.0 | 7.9 | 8.1 | 8.5 | 8.2 | 9.9 |
| 18 | $A_3B_3C_2$ | 7.5 | 19 | 20.4 | 9.9 | 9.6 | 10 | 9.8 | −7.7 |
| 19 | $A_1B_1C_3$ | 3.75 | 5 | 39.0 | 7.7 | 7.5 | 7.5 | 7.6 | 16.5 |
| 20 | $A_2B_1C_3$ | 3.75 | 7 | 34.5 | 7.7 | 7.2 | 7.4 | 7.4 | 18.7 |
| 21 | $A_3B_1C_3$ | 3.75 | 9 | 30.6 | 9.3 | 9.3 | 9.2 | 9.3 | −2.2 |
| 22 | $A_1B_2C_3$ | 7.5 | 5 | 39.0 | 6.9 | 7.3 | 6.9 | 7 | 23.1 |
| 23 | $A_2B_2C_3$ | 7.5 | 7 | 34.5 | 6.4 | 6.6 | 6.7 | 6.6 | 27.5 |
| 24 | $A_3B_2C_3$ | 7.5 | 9 | 30.6 | 7.8 | 7.8 | 7.4 | 7.7 | 15.4 |
| 25 | $A_1B_3C_3$ | 11.25 | 5 | 39.0 | 9.8 | 10 | 9.5 | 9.8 | −7.7 |
| 26 | $A_2B_3C_3$ | 11.25 | 7 | 34.5 | 9 | 9.1 | 9 | 9 | 1.1 |
| 27 | $A_3B_3C_3$ | 11.25 | 9 | 30.6 | 10.2 | 9.7 | 10.5 | 10.1 | −11 |

*A*: disc coverage ratio, *B*: dome height to diameter ratio, *C*: dome base diameter, h: dome height, n: the approximate number of domes existing on the disc, L: dome-to-dome spacing, and Δ: change in drag force of domed disc with respect to flat disc, such that negative values of "Δ" indicate a deterioration.

*2.4. Design of Experiment and Statistics*

This study was conducted in two distinct stages. Stage one aimed to determine the optimal levels of the control parameters mentioned above that produce the minimum possible drag force under paddy field conditions. During this stage, the silty loam soil "S1" with a moisture content of 30% on a dry weight basis was used. Other than the control, 27 tests were conducted with three replications according to the L27 ($3^3$) Taguchi orthogonal array using all possible combinations of control parameter levels mentioned in Table 2. The optimal levels of the control parameters were searched using signal-to-noise ratio analysis (S/N). Typically, there are three strategies to analyze the S/N ratio: lower is better, higher is better, and nominal is better [26]. In this experiment, lower drag force values were desired. Therefore, a lower-is-better strategy was adopted and calculated using Equation (3) [21].

$$SNR = -10\log\left[\frac{1}{n}\sum_{i=1}^{n} y_i^2\right] \tag{3}$$

Here, *n* is the number of all trials and $y_i$ is the measured data of trial number $i^{th}$.

The individual contributions from all control parameters in the drag force were determined by utilizing analysis of variance (ANOVA) with a 5% significance level. Minitab 18 software was used to graphically depict the relationships between control parameters and measured values of drag force using the contour plot in two dimensions.

Stage two entailed comparing the sliding resistance of the optimized domed disc (defined in stage one) with that of the flat disc under different soil conditions (soil textures

of silty loam and sandy clay loam at three moisture contents of 23, 30, and 37%). The significance of the comparisons was verified via paired-samples *t*-test. Finally, the experimentally achieved drag force of the optimized disc was compared with that of the flat disc under all treatments according to the following equation:

$$\Delta\% = \frac{\tau_F - \tau_B}{\tau_F} \times 100 \qquad (4)$$

where $\Delta$ is change in drag force of optimized disc with respect to flat disc in similar soil conditions, and $\tau_F$ and $\tau_B$ are the mean drag force values of the flat disc and the optimized disc, respectively.

## 3. Results

### 3.1. The Exploratory Experiment Findings

According to an L9 Taguchi orthogonal array (discs numbered 1, 6, 8, 11, 13, 18, 21, 23, and 25), a preliminary test was performed to cross-check the rationality of the selected levels of control parameters. The data resulting from this preliminary test are visually displayed in a two-dimensional contour plot in Figure 5. The lowest values of drag force (bright spots) were observed near the center of each graph. This, in turn, may demonstrate the rationality of the selected values. The contour plot shows that the lowest values of drag force (bright spots) were associated with the HDR range of 15% to 30%, indicating that this range had the greatest impact on the drag force reduction.

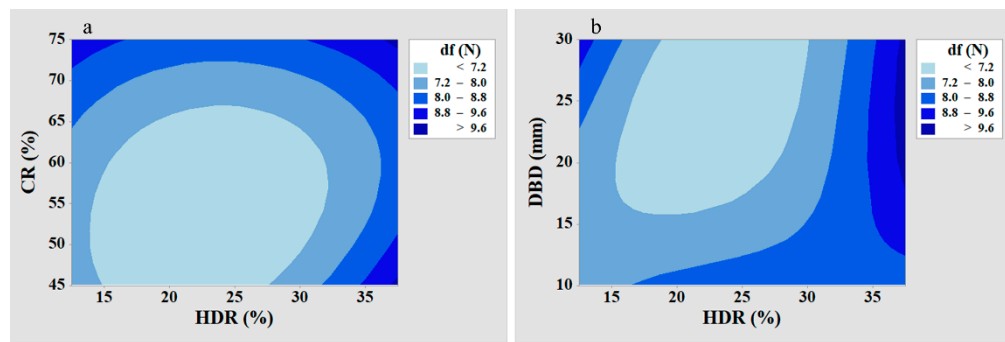

**Figure 5.** Effect of control parameters on sliding resistance according to an L9 ($3^3$) Taguchi orthogonal array: (**a**) contour plot of sliding resistance vs. CR, HDR; (**b**) contour plot of sliding resistance vs. DBD, HDR.

### 3.2. Full Factorial Experiment Results

The main experiment was conducted according to an L27 ($3^3$) full factorial design with three replications. Table 3 shows the mean drag force along each disc stroke and the geometrical specifications of the various discs. The minimum drag force was 6.6 N with the $A_2B_2C_3$ parameter combination (Disc 23). In comparison, the maximum drag force was 10.9 N with the $A_3B_3C_1$ parameter combination (Disc 9). The highest achieved difference in the measured values of the replications was 0.8 N, indicating that the test results utilizing the testing method described in this study were repeatable (Figure 6).

The results revealed that domed discs with a dome height-to-diameter ratio of 25% had reduced drag force of up to 27.5% (Disc 23) compared to a flat disc, while domed discs with a dome height-to-diameter ratio of 37.5% had increased drag force of up to 20% (Disc 9). In contrast, Soni and Salokhe [23] reported that domed surfaces with a dome height-to-diameter ratio of 50% or less had decreased drag force of up to 30%. This contrast in results may be related to the different experimental conditions used in both experiments where Soni and Salokhe [23] used heavy clay soils (62% clay) with liquid and plastic limits of 48% and 28.9%, respectively, unlike the experimental conditions used in the current study (Table 2). Furthermore, they used high levels of soil moisture content, reaching 60.1% (d.b.).

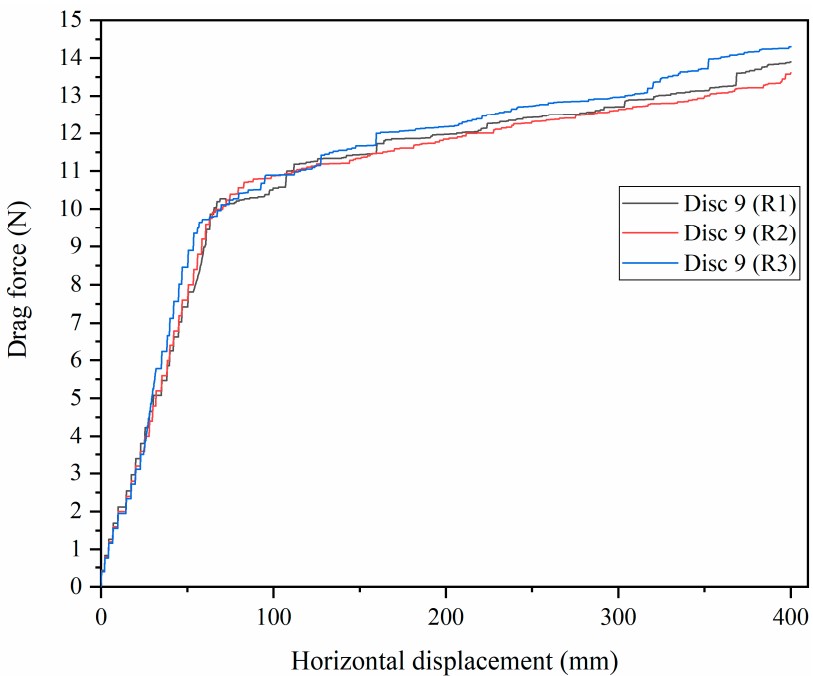

**Figure 6.** Repeatability of tests (Disc 9, silty loam soil "S1" with a moisture content of 30%).

### 3.2.1. Analysis of Variance (ANOVA) for Drag Force (df)

Analysis of variance with a 95% confidence level was utilized in this study to identify the individual influences of the control parameters on the drag force (response values). The ANOVA results for drag force are provided in Table 4. The last column of the table shows the percentage value of each parameter contribution, which reflects the degree of influence on the drag force values. According to Table 4, the percentage contributions of the A, B, and C parameters on the drag forces were determined to be 29.4%, 58.5%, and 1.6%, respectively. Thus, the dome height-to-diameter ratio was the most dominant factor affecting drag forces. The dome height-to-diameter ratio importance stems from the fact that it accommodates the vertical and lateral dimensions of the dome, directly affecting the attached area of the interfacial water film. The disc coverage ratio (factor A, 29.4%) also showed a considerable contribution. In contrast, the dome base diameter (factor C, 1.6%) had limited influence on drag force. The mean square of the term "HDR × DBD" was less than the mean square of error; that is why "HDR × DBD" was embedded in the error term, and the corrected error $e^\Delta$ was thus obtained. The percentage contribution of the corrected error $e^\Delta$ for sliding resistance was within the acceptable limit (less than 20%) [27].

**Table 4.** Analysis of variance for drag force of test discs.

| Variance Source | DF | Sum of Squares | Mean Square | F-Value | *p*-Value | Contribution/% |
|---|---|---|---|---|---|---|
| CR | 2 | 11.80 | 5.90 | 77.49 | 0.000 | 29.4 |
| HDR | 2 | 23.53 | 11.76 | 154.55 | 0.000 | 58.5 |
| DBD | 2 | 0.63 | 0.31 | 4.12 | 0.043 | 1.6 |
| CR × HDR | 4 | 2.18 | 0.55 | 7.17 | 0.003 | 5.4 |
| CR × DBD | 4 | 1.14 | 0.28 | 3.74 | 0.034 | 2.8 |
| HDR × DBD $\Big\}e^\Delta$ | 4 $\Big\}$12 | 0.38 $\Big\}$0.91 | 0.08 | | | 2.3 |
| Error | 8 | 0.53 | | | | |
| Total | 26 | 40.18 | | | | 100 |

### 3.2.2. Analysis of the S/N Ratio for Drag Force (df)

Based on the L27 ($3^3$) full factorial design results, the S/N ratios were computed as per the "lower is better" strategy and are presented in Table 5. The S/N ratio analysis showed

that the highest S/N ratio values were related to the second level of each experimental parameter, indicating that the second level of each experimental parameter had the greatest effect on drag force reduction. Accordingly, the optimal disc was determined as $A_2B_2C_2$ (Disc 14), i.e., disc coverage ratio of 60%, dome height-to-diameter ratio of 25%, and dome base diameter of 20 mm. In contrast, the measured drag force values indicated that Disc 23 had the lowest sliding resistance over the whole range of the current experiment. This discrepancy can be explained by understanding the Taguchi technique's working methodology, which determines the optimal level of each experimental parameter by isolating the direct effects of experimental parameter levels on response values while ignoring the influence of uncontrollable factors. Thereby, the small drop in drag force of Disc 23 compared to Disc 14 (0.2 N) can be attributed to uncontrollable factors such as a tiny fluctuation in soil paste compaction as a result of surface flattening following each run, as well as a slight decline in soil moisture content during the experiments. Based on the foregoing, Disc 14 was selected as the optimal disc, and it was subsequently tested under various soil conditions.

**Table 5.** Signal-to-noise response table for drag force.

| Level | Control Parameters | | |
|:---:|:---:|:---:|:---:|
| | CR (%) | HDR (%) | DBD (mm) |
| 1 | −17.95 | −18.14 | −18.38 |
| 2 | −17.56 | −17.07 | −18.02 |
| 3 | −19.15 | −19.46 | −18.27 |
| Delta | 1.59 | 2.39 | 0.36 |
| Order | | HDR > CR > DBD | |

### 3.3. Confirmation of Experiment Results

The purpose of the confirmation experiments was to determine whether the biomimetic effect of the optimized disc could be extended to a variety of soil conditions. The drag force data for all treatments are summarized and analyzed in Table 6. The results showed that, as compared to the flat disc, the optimized biomimetic disc could improve drag force by 9 to 25%, depending on the soil condition. According to the paired samples t-test with a 95% confidence level, there were significant differences in drag force between the flat disc and the optimized biomimetic disc in all treatments ($p < 0.05$). In addition, the statistics indicated that variations in soil moisture content could affect the soil-to-disc drag force by up to 35% in the range of this experiment. Based on the least significant difference (LSD) test results, statistically different treatments are highlighted in Figure 7.

**Table 6.** Statistical analysis on the drag force of the flat disc against that of optimized biomimetic disc.

| Test Condition | Disc Type | Mean $\tau$ (N) | Std. Dev | t Value | *p* Value | Δ (%) |
|:---:|:---:|:---:|:---:|:---:|:---:|:---:|
| 23% MC, S1 | Flat | 8.3 | 0.37 | 9.67 | 0.001 | 18.1 |
| | Biomimetic | 6.8 | 0.43 | | | |
| 30% MC, S1 | Flat | 9.4 | 0.50 | 5.67 | 0.005 | 24.5 |
| | Biomimetic | 7.1 | 0.37 | | | |
| 37% MC, S1 | Flat | 10.5 | 0.57 | 16.65 | <0.001 | 21.9 |
| | Biomimetic | 8.2 | 0.53 | | | |
| 23% MC, S2 | Flat | 6.9 | 0.40 | 8.45 | 0.001 | 23.2 |
| | Biomimetic | 5.3 | 0.20 | | | |
| 30% MC, S2 | Flat | 8.8 | 0.37 | 26.70 | <0.001 | 25.0 |
| | Biomimetic | 6.6 | 0.43 | | | |
| 37% MC, S2 | Flat | 6.5 | 0.40 | 3.36 | 0.028 | 9.2 |
| | Biomimetic | 5.9 | 0.17 | | | |

S1: silty loam soil, S2: sandy clay loam soil, optimized biomimetic disc: Disc 14, Std. Dev: standard deviation, Δ: change in drag force of optimized disc with respect to flat disc in similar soil conditions. The paired-samples *t*-test was conducted using five runs for each soil condition.

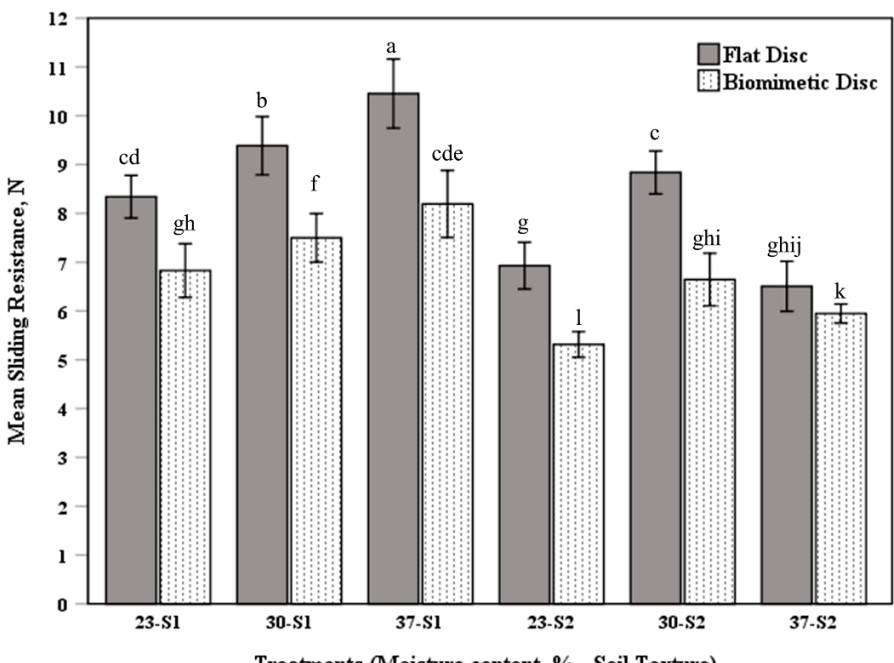

**Figure 7.** Comparing the mean drag force of the flat surface with that of the optimized biomimetic surface in all treatments. Means with different letters are significantly different.

## 4. Discussion

Statistics revealed that the dome height-to-diameter ratio and disc coverage ratio were crucial parameters that influence the drag force of domed surfaces, while the dome base diameter had limited influence. The dome height-to-diameter ratio was arguably the most dominant parameter, with a percentage contribution of 58.5%. The dominance of the dome height-to-diameter ratio was mainly due to the close linkage between the capacity of domes to break the continuity of the interfacial water film and the extent of dome convexity [28,29]. Moreover, the dome height-to-diameter ratio affected the extent of disc sinkage into soil paste, hence, soil resistance on the disc's front side. The disc coverage ratio also showed a significant contribution (29.4%) to the disc drag force. The results showed that the higher the disc coverage ratio, the more disc drag force. At a coverage ratio greater than 60%, the increase in drag force could be attributed to a blockage of soil movement between adjacent domes, causing the soil to accumulate in front of the disc along the disc stroke.

Results revealed that the lowest drag force values were associated with the HDR range of 15% to 30%, while domed discs with a dome height-to-diameter ratio of 37.5% had increased drag force of up to 20% (Disc 9) compared to a flat disc. In contrast, Soni and Salokhe [23] reported that domed surfaces with a dome height-to-diameter ratio of 50% or less had decreased drag force. This contrast in results may be related to the different soil textures used in both experiments, since Soni and Salokhe [23] used heavy clay soil with 62% clay, whereas the current study used silty loam and sandy clay loam soils with 15% and 21% clay, respectively. According to Srivastava et al. [10], the fine soil particles (clay and fine silt) have a higher capacity for water-holding due to their large surface area; thus, heavy clay soil is stickier than silty loam and sandy clay loam soils.

The optimized domed disc (Disc 14) successfully reduced drag force compared to the flat disc under varying soil moisture contents ranging from 23% to 37%. The maximum recorded drag force reduction was 25% in the silty loam soil with 30% moisture content. The reduced drag force of Disc 14 could be attributed to the ability of domes to push the soil away from concave areas between adjacent domes while dragging, restricting interfacial water film continuity. Another reason for the drag force reduction by the optimized domed disc was air retention in the concave areas between adjacent domes, which provided a gas isolation layer, reducing soil-disc adhesion. Furthermore, the tangential movement of the

optimized domed disc caused disturbances in the surrounding soil layer, resulting in a decrease in the soil–disc friction coefficient [30].

The drag force test in silty loam soil (S1) revealed that raising the soil moisture content below the liquid limit increased drag force. This increase in drag force associated with the increase in soil moisture content was probably due to the formation of a thin interfacial water film between the disc and the soil, increasing soil–disc adhesion. The soil–disc adhesion, therefore, acted as an extra load, resulting in increased drag force. On the other hand, the expected increase in initial disc subsidence due to rising soil moisture content could increase soil resistance on the disc's front side, hence drag force. In comparison, the soil–disc sliding resistance in sandy clay loam soil (S2) followed a distinct pattern. The soil–disc sliding resistance was initially low at the lowest moisture content (23%); as the moisture content increased to 30%, the soil–disc sliding resistance increased. Finally, as the moisture content increased even further, the soil–disc sliding resistance decreased. The marked difference in sensitivity to changes in soil moisture content between S1 and S2 was probably attributable to differences in fine soil particle content (clay and fine silt). In contrast to coarse soil particles, fine soil particles have a high water-holding capacity due to their large surface area and chemical interactions. Due to the lower fine soil particle content of S2, the sliding resistance was reduced as the moisture content reached the liquid limit due to the lubricating effect generated by the interfacial free water layer [8].

## 5. Conclusions

The micro-convex structure of the dung beetle head was used as a biomimetic prototype to create a number of domed discs in order to examine the influence of specific geometrical parameters on sliding resistance reduction using an L27 ($3^3$) Taguchi orthogonal array. Variance analysis (ANOVA) revealed that the dome height-to-diameter ratio was the most dominant parameter, followed by the disc coverage ratio. In contrast, the dome base diameter had a limited influence on drag force. Using the signal-to-noise ratio analysis per the lower-is-better strategy, the optimal disc for minimizing drag force was determined to be Disc 14, with a disc coverage ratio of 60%, a dome height of 5 mm, and a dome base diameter of 20 mm.

The sliding resistance of the optimized domed disc (Disc 14) and that of the flat disc were compared under certain soil conditions (soil textures of silty loam and sandy clay loam; soil moisture contents of 23%, 30%, and 37%) to investigate the biomimetic effect of the optimized domed disc. The optimized domed disc produced less sliding resistance than the flat disc in all treatments, by around 9% to 25%, depending on the soil conditions. The results obtained in this experimental study can be used to support the manufacture of paddy soil–engaging components such as flattening tools and fertilizer furrow openers.

**Author Contributions:** Conceptualization, A.E.S. and G.Z.; methodology, A.E.S. and H.W.; software, A.E.S. and Y.G.; validation, A.E.S. and X.Z.; formal analysis, A.E.S. and M.A.A.; investigation, A.E.S.; resources, M.A.A.; data curation, H.W.; writing—original draft preparation, A.E.S.; writing—review and editing, A.E.S. and M.A.A.; visualization, X.Z. and H.W.; supervision, G.Z.; project administration, G.Z.; funding acquisition, G.Z. All authors have read and agreed to the published version of the manuscript.

**Funding:** This study was financially supported by the National Natural Science Foundation of China; (Grant No. 51775220) titled "Contract–Rheology-Resistance coupling mechanism between the flat plate type soil working parts and disturbed paddy soil".

**Institutional Review Board Statement:** Not applicable.

**Informed Consent Statement:** Not applicable.

**Data Availability Statement:** The data presented in this study are available on-demand from the first author at (abouelnadar@webmail.hzau.edu.cn).

**Acknowledgments:** The authors would like to express their gratitude to the Huazhong Agricultural University soil physics laboratory staff for their assistance with the laboratory experiments.

**Conflicts of Interest:** The authors declare no conflict of interest.

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
