# Peer review of "Effect of Biomimetic Surface Geometry, Soil Texture, and Soil Moisture Content on the Drag Force of Soil-Touching Parts"

_applsci, doi:10.3390/app11198927_

Round 1

Reviewer 1 Report

  1. It is not quite clear what Figure 1 is for. Especially the second part of the figure is not clear.
  2. Domed disc were made from acrylonitrile butadiene styrene. In the field would a disk of this material also be used? How resistant is it to sunlight?
  3. What was the rationale for the choice of humidity values (23, 30, and 37%)? What is the moisture content of the rice fields?
  4. At what stage of rice cultivation is dome disc treatment used?

Reviewer 2 Report

Dear Authors, you can find some comments and suggestion in the pdf file. The research activity seems to be interesting, however it is necessary to better support the results attained in order to clarify the "optimized disc choice" methodology. At present it seems that some paramenters are not so deeply investigated. I suggest to review the proposed results and support them with other comparative tests. Finally it should be useful to figure out how the shape of the optimized disc may affect the "real" working condition of some tools/interchangeable equipment.

Author Response

Dear reviewer,

Reviewer 3 Report

Interesting study and well conducted tests, I suggest to improve the quality of the discussion and introduction following the comments in the attached pdf file

Round 2

Reviewer 2 Report

Dear Authors, thank you for your effort for improving the paper and giving an answer to proposed comments. All the comments are resolved and the paper results to be complete and clear.

Just in line 335 you should remove brackets for Disc 14.

Author Response

Dear reviewer,

Reviewer 3 Report

First, the modifications which authors performed in the second version are not highlighted in track change (as clearly stated in the guide for authors of the journal) making the review process difficult. This is not something which has negative effects on the  quality of the manuscript but highlights that the authors carried out the review process in a sloppy way and this is not a positive feature.

Moreover, my previous comments were only partially addressed, mostly regarding the issue of improving the discussion section. I therefore suggest a further step of review also in this case asking the authors to improve the discussion section of the manuscript...

Author Response

Dear reviewer,

Round 3

Reviewer 3 Report

Accepted 

Author Response

Dear reviewer,
